# SUBGRAPH MINING FOR GRAPH NEURAL NETWORKS

## ABSTRACT

While Graph Neural Networks (GNNs) are state-of-the-art models for graph learning, they are only as expressive as the basic first-order Weisfeiler-Leman graph isomorphism test algorithm. To enhance their expressiveness one can incorporate complex structural information as attributes of the nodes in input graphs. However, this approach typically demands significant human effort and specialised domain knowledge, which is not always available. In this paper, we demonstrate the feasibility of automatically extracting such structural information through subgraph mining and feature selection techniques. Our extensive experimental evaluation, conducted across graph classification tasks, reveals that GNNs extended with automatically selected features obtained using subgraph mining can achieve comparable or even superior performance to GNNs relying on manually crafted features.

## 1 INTRODUCTION

Graphs as abstract mathematical structures can represent relational data such as biological compounds and social networks. As such, numerous graph-based learning methodologies have been proposed to address tasks such as comparative analysis, classification, regression, etc. Broadly, graph learning methodologies can be categorised into two main domains: sub(graph) mining (Cheng et al., 2014) and deep learning approaches (Scarselli et al., 2008; Jegelka, 2022). Unfortunately, the existing approaches to graph learning have limitations in terms of scalability, interpretability, and expressivity.

*Graph mining*, also known as traditional graph learning, relies mainly on hand-crafted features as a signal to distinguish and analyse graphs between each other. There exist multiple graph mining techniques (Cheng et al., 2014; Ribeiro et al., 2021), such as gSpan (Yan & Han, 2002) and Gaston (Nijssen & Kok, 2005). These techniques are interpretable, however, they typically have high computational complexity due to involving the subgraph isomorphism problem as a sub-routine (Ribeiro et al., 2021).

*Deep learning* techniques such as Graph Neural Networks (GNNs) are state-of-the-art for graph learning. With these methods features of graphs can be learned automatically in an end-to-end fashion. However, there are two problems with GNNs. First, they are of a black-box nature, lacking interpretability. Second, there is a trade-off between complexity and expressivity, as standard GNNs are as expressive as the 1-WL isomorphism test (Leman & Weisfeiler, 1968; Huang & Villar, 2021) and higher-order GNNs with increased expressivity come at a computational cost (Morris et al., 2019; Jegelka, 2022). The most popular framework based on deep learning for graph learning is known as Message Passing Neural Networks (MPNNs) (Gilmer et al., 2017). MPNNs are a very practical approach for large graphs since the memory complexity increases linearly with the graph size making them very desirable. However, they fall into the problem of limited expressivity as shown by Xu *et. al.* (Xu et al., 2018; Jegelka, 2022). As a consequence, GNNs based on MPNNs, fail to detect certain graph structures such as *cycles* or *cliques*.

In recent years, to address the problem of limited expressiveness while keeping the computational cost feasible, researchers have introduced MPNN-based approaches capable of capturing higher-order graph structures. To this end, one approach encodes subgraph information directly in the message-passing process. Relevant to this paper are Graph Substructure Networks (GSN) (Bouritsas et al., 2022), where subgraph occurrences are used to enhance the feature representation of the input graph.

To address these challenges, we propose an automated framework bridging the gap between traditional graph and deep learning. Specifically, we leverage subgraph mining techniques to generate substructures that enhance the performance of substructure-based MPNNs. Our approach selects various off-the-shelf subgraph mining techniques to generate a set of candidate substructures for performance improvement. Furthermore, we employ different feature selection strategies to identify the most useful substructures. The selected substructures are then encoded at the node or edge level for the given graphs. To summarise, this work makes the following contributions: 1) We propose an automated framework to select substructures for use with substructure-based GNNs; 2) We show that our framework can be used to automatically select 'useful' substructures to use with subgraph-based MPNNs; 3) We perform an extensive evaluation on graph classification tasks. The paper is structured as follows: In section 2 we introduce notions and definitions used throughout the paper. Next, in section 3 our proposed framework is described. Section 4 includes experiments carried out to compare our proposed methodology with existing approaches. Section 5 discusses the related work. Section 6 concludes the work and describes potential future directions.

## 2 PRELIMINARIES

In this section, we introduce concepts and terminology. When dealing with graph data, we distinguish between two types: a single large graph $G = (V, E)$ and a set of graphs $D = \{G_1, G_2, \ldots, G_n\}$. In the context of a single large graph, we may also use *network*. Similarly, terms *subgraph* and *substructure* are used interchangeably.

**Graph** A graph $G = (V, E)$ consists of a set $V(G)$ of vertices or nodes, and a set $E(G)$ of edges. The set of nodes in a graph represents entities, while the set of edges represents binary relations between nodes. Edges are represented as tuples in the form $(u, v)$, where $u, v \in V(G)$. Additionally, the nodes and edges can be associated with labels using a labelling function $l$, i.e. $l(u)$ or $l(u, v)$. Graphs can be directed, where the set of edges consists of ordered pairs $(u \rightarrow v)$, or undirected, where edges have no direction $(u \leftrightarrow v)$. Furthermore, in a graph, a node can be assigned a feature vector $x_v$ for $v \in V(G)$ and an edge a edge feature vector $x_e$ for $e \in E(G)$.

**Graph Isomorphism** Given two graphs $G = (V, E)$ and $H = (V, E)$, $G$ is isomorphic to $H$ if they are structurally equivalent. More formally, if there exists a bijective function $f : V(G) \rightarrow V(H)$ such that for any edge $(u, v) \in E(G)$ there is a corresponding edge $(f(u), f(v)) \in E(H)$ and vice versa. Two isomorphic graphs are denoted by $H \simeq G$. In the case of labelled graphs, where nodes, edges, or both nodes and edges can have labels, the bijective function must take into account the labels as well. Specifically, given two graphs and their labelling functions $(l_v(V(G)), l_e(E(G)))$ and $(l'_v(V(H)), l'_e(E(H)))$, $G$ and $H$ are isomorphic labelled graphs if there is a bijective function $f : V(G) \rightarrow V(H)$ such that for any edge $(u, v) \in E(G)$ there is a corresponding edge $(f(u), f(v)) \in E(H)$ and $\forall v \in V(G) : l_v(v) = l'_v(f(v))$ and $\forall (u, v) \in E(G) : l_e(u, v) = l'_e(f(u), f(v))$.

**Subgraph** A subgraph $S = (V, E)$ of a graph $G = (V, E)$ is a graph that is obtained by selecting a subset of nodes and edges from $G$ such that $V(S) \subseteq V(G)$ and $E(S) \subseteq E(G)$. An induced subgraph $S = (V, E)$ of $G$ is a graph obtained from a subset of the vertices $G$ and all of the edges of $G$ connecting pairs of vertices in that subset. More formally, $\forall (v, u) \in V(S) : (v, u) \in E(G) \leftrightarrow (v, u) \in E(S)$.

**Subgraph Isomorphism** Subgraph isomorphism is the problem of deciding, given two graphs $G$ and $S$, whether $G$ contains a subgraph $G_k$ that is isomorphic to $S$, i.e. $G_k \simeq S$. In the case of labelled graphs, the subgraph isomorphism must consider labels of nodes and edges as in the case of graph isomorphism. The subgraph isomorphism problem is known to be NP-Complete (Cheng et al., 2014).

**Graph Neural Networks** In GNNs, there are three tasks: The first task is *Node prediction*, where each node $v \in V$ has a label $y_v$ and the goal is to learn an embedding $h_v$ of $v$ such that we can use to predict $v$'s label as $y_v = f(h_v)$. The second task *Edge prediction*, where each edge $e \in E$ has a label $y_e$ and the goal is to learn an embedding $h_e$ of $e$ such that we can use to predict $e$'s label as $y_e = f(h_e)$. The third task is *Graph prediction*, where we are given a set of graph $G = \{G_1, G_2, \ldots, G_N\}$ and their labels $Y = \{y_1, y_2, \ldots, y_n\}$, and the goal is to learn an embedding $h_G$ such that we can use to predict an entire graph $y_G = f(H_G)$. Message Passing Neural Networks (MPNNs) are a general framework for GNNs. Specifically, node-to-node information is propagated by iteratively

aggregating neighbouring node information to a central node. After $k$ iterations of aggregation, a node's representation captures the structural information within its k-hop neighbourhood. Formally, the $k$-th layer of an MPNN is:

$$a_v^{(k)} = AGGREGATE^{(k)}(\{h_u^{(k-1)} : u \in \mathcal{N}(v)\}), \tag{1}$$

$$h_v^{(k)} = COMBINE^{(k)}(h_v^{k-1}, a_v^k) \tag{2}$$

where $h_v^{(k)}$ represents the feature vector of node $v$ at the $k$-th iteration and $\mathcal{N}(v)$ represents all the neighbouring nodes of node $v$. For node classification, $h_v^{(K)}$ of the final iteration is used for prediction, while for graph classification, the READOUT function is used to obtain the graph's representation $h_G$:

$$h_G = READOUT(\{h_v^{(K)} \mid v \in G\}). \tag{3}$$

Details on how to choose $AGGREGATE^{(k)}(\cdot)$, $COMBINE^{(k)}(\cdot)$, and $READOUT(\cdot)$ can be found at (Xu et al., 2018).

**Weisfeiler-Lehman (WL) tests** This test also known as "naïve vertex refinement" is a fast heuristic to check whether two graphs are isomorphic or not. The *Weisfeiler-Lehman (WL) test* (Leman & Weisfeiler, 1968) and its 1-dimensional form (1-WL), are analogous to neighbour aggregation (message-passing) in Graph Neural Networks (Xu et al., 2018; Huang & Villar, 2021; Bouritsas et al., 2022). The WL test iteratively aggregates the labels of nodes and their neighbourhoods, and it hashes the aggregated labels into unique new labels. For a given undirected graph $G = (V, E, X_v)$, where $X_v$ is the set of node features, a hash for every node is provided:

$$h_v^k = \text{HASH}(h_v^{k-1}, \{\{h_u^{k-1} : u \in \mathcal{N}(v)\}\}) \, \forall v \in V, \tag{4}$$

where $h_v^k$ represents the feature vector of node $v$ at the $k$-th iteration, and $\mathcal{N}(v)$ denotes the neighbours of a node $v$. The algorithm decides that the two graphs are non-isomorphic if the hashed labels of the nodes between the two graphs differ at some iteration. Most of the work done on improving GNN expressivity is based on mimicking the higher-order generalisation of WL, known as $k$-WL and $k$-Folklore WL. The main idea of higher-order WL is to aggregate neighbouring information in $k$-nodes instead of a single neighbouring node.

**Substructure-based GNNs** GNNs capable of making use of more informative topological features have been proposed as is Graph Substructure Network (GSN) (Bouritsas et al., 2022). GSNs make minimal changes to the basic MPNN architecture by enriching the message passing with more complex predefined structural features, such as the number of cliques or cycles to which nodes or edges belong. The message passing of GSNs is defined as follows:

$$h_v^{(k+1)} = COMBINE^{(k+1)}(h_v^k, a_v^{k+1}), \tag{5}$$

$$a_v^{(k+1)} = \begin{cases} A^{(k+1)}(\{h_v^{(k)}, h_u^{(k)}, \mathbf{x}_v^V, \mathbf{x}_u^V, \mathbf{e}_{u,v} : u \in \mathcal{N}(v)\}) \text{ (GSN-v)} \\ A^{(k+1)}(\{h_v^{(k)}, h_u^{(k)}, \mathbf{x}_{u,v}^E, \mathbf{e}_{u,v} : u \in \mathcal{N}(v)\}) \text{ (GSN-e)} \end{cases} \tag{6}$$

where $\mathbf{x}_v^V$ and $\mathbf{x}_{u,v}^E$ are the additional features obtained from the provided topological features counts (e.g. number of cliques or cycles).

## 3 AUTOGSN

We propose AUTOGSN, a framework for increasing the expressivity of GNNs by augmenting the feature representation of the input graphs with automatically discovered subgraph patterns. Similarly to GSN (Bouritsas et al., 2022), AUTOGSN encodes substructural information in the message passing of graph neural networks, by enriching node or edge representations. However, AUTOGSN automates the substructure selection, alleviating the need for human knowledge and allowing the discovery of unknown informative structural features. AUTOGSN operates as a hyperparameter search that optimises a set of mining and feature selection tools, instead of directly searching over the set of possible structural features, as in (Bouritsas et al., 2022). AUTOGSN is available at `https://anonymous.4open.science/r/AutoRFE-171C`.

In this section, we describe the different steps required to extract and augment the input graphs, together with the choices of mining and selection tools. A high-level overview of the flow of AUTOGSN is described in Algorithm 1, and illustrated in Figure 1.

---

**Algorithm 1** AUTOGSN

---

**Input:**
   $D_t, D_v$                                    ▷ The training and validation sets of graphs
**Output:** $score$            ▷ A score for the current subgraph mining and selection strategy
    $S = MINING(D_t)$
    $S' = SELECTION(S)$
    $D_t^{aug} = COUNTING(D_t, S')$
    $D_v^{aug} = COUNTING(D_v, S')$
    $GNN.train(D_t^{aug})$
    $score = GNN.evaluate(D_v^{aug})$

---

Figure 1: Graphical overview of the AUTOGSN framework. A set of graphs (*Dataset*) are provided as input for different mining systems (*Mining*), which generate a set of subgraphs. Next, selection strategies (*Selection*) are applied to select relevant subgraphs with respect to the classification task. Lastly, the subgraphs are encoded in node or edge level (*Counting*). The augmented dataset is then used for downstream tasks.

## 3.1 SUBGRAPH MINING

The initial step in AUTOGSN involves subgraph generation, i.e. extracting a set of informative subgraphs $S = \{H_1, H_2, ..., H_m\}$ from an input graph dataset $D = \{G_1, G_2, ..., G_n\}$. To this end, AUTOGSN employs off-the-shelf subgraph mining techniques. Clearly, there are different criteria to determine which subgraphs are informative, which leads us to investigate three classes of mining techniques: *(i)* frequent subgraph mining, *(ii)* discriminative subgraph mining and *(iii)* rule mining.

*Frequent subgraph mining (FSM)* systems, such as GSPAN (Yan & Han, 2002) and GASTON (Nijssen & Kok, 2005), compute all connected frequent subgraphs within a graph database. Specifically, when provided with a set of graphs $D$ and a minimum support threshold $minSup$, FSM identifies all graphs $S_D$ that are subgraphs in at least $minSup$ of the graphs in $D$. They are often used as features to perform classification or regression. One downside of frequent subgraph mining systems is that they exclusively work with fully labelled graphs, for example, molecular graphs with labelled nodes and edges. This limitation is a result of complexity that arises when subgraph isomorphism is executed over unlabelled graphs.

*Discriminative subgraph mining systems*, like GAIA (Jin et al., 2010), work on a set of graphs $D$ that are classified and that can be used to discriminate between two or more classes. This is usually formulated in terms of scoring higher than a threshold w.r.t. a criterion such as entropy or $\chi^2$, or alternatively, finding the top-$k$ best scoring patterns. Discriminative frequent subgraph mining being an extension of FSMs systems only operate on fully labelled graphs for the same reasons of rising complexity with unlabelled subgraph isomorphism.

*Rule mining systems*, including ANYBURL (Meilicke et al., 2019) and RDFRULES (Zeman et al., 2021), have been proposed in the knowledge graph community. They focus on learning to complete a partial knowledge graph by learning *if-then* rules, like:

$$h \leftarrow b_1, b_2, ..., b_n$$

which states that the head $h$ must be in the knowledge graph if all the body elements $b_1, b_2, ..., b_n$ are in the graph. In the context of knowledge graphs, both $h$ and $b_i$ are binary predicates of the form $r(X, Y)$, where $r$ is the relation type and $X$ and $Y$ are variables, i.e. placeholders for entities. An

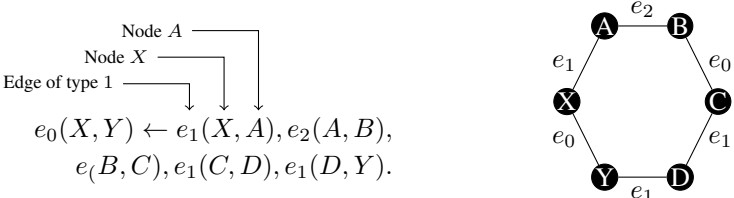

Figure 2: AUTOGSN — Conversion of *if-then* rules to graphs.

example of an if-then rule is shown in Figure 2 (left). In order to adapt rule miners for knowledge graphs to our subgraph generation task, we go through the following steps. First, we build a unique big knowledge graph by the union of single graphs contained in the original dataset. In particular, $KG = (N_{kg}, E_{kg})$, where $N_{kg} = N(G_1) \cup ... \cup N(G_n)$ and $E_{kg} = E(G_1) \cup ... \cup E(G_n))$. Second, after mining the rules, we translate them into a set of subgraphs. Given a rule $\rho$, let $\mathcal{X}$ be the set of all variables in the rule and let $\mathcal{R} = \{h\} \cup \{b_1, b_2, ..., b_n\}$. We translate each rule $\rho$ into a subgraph $S = (\mathcal{X}, \mathcal{R})$, where the variables are turned into the nodes of the subgraph and the (binary) relations into its edges. An illustration of this process is shown in Figure 2. Finally, we need to solve a final discrepancy between subgraph mining techniques and rule mining ones. Subgraph mining techniques are designed to incrementally expand subgraphs (e.g., starting from an edge and progressing to a triangle), thus generating subgraphs that are not isomorphic per construction. However, rule mining techniques may generate rules that correspond to isomorphic subgraphs. To avoid such repeated subgraphs, we *filter* the generated rules such that only unique graphs are kept. We compare all the generated graphs through the VF2 algorithm (Cordella et al., 2004). While such filtering adds an extra level of complexity, it saves from counting identical subgraphs multiple times in the counting step (see Section 3.3)

### 3.2 SELECTION

Controlling the number of generated subgraphs is fundamental because our final goal is to count how many occurrences of the mined subgraphs are present in the input graphs and use this count to augment the graph representation (see Section 3.3). However, the output of the mining process is in general a large set of subgraphs, leading to two issues. First, counting may become computationally unfeasible, due to the need to repeatedly call subgraph isomorphism routines for each of the mined subgraphs. Second, encoding the counts of a large number of generated subgraphs introduces a lot of redundancy (this is particularly true for features extracted by frequent subgraph mining techniques). To avoid these two problems, in this step, we select only a subset of the generated graphs using feature selection strategies. To this end, we turn each input graph $G \in D$ into a table by employing a boolean encoding of the generated subgraphs $S = \{S_1, ..., S_m\}$. In particular, for each graph $G$, we build a binary vector $x \in \{0, 1\}^m$, such that $x_i = 1$ if $H_i$ is a subgraph of $G$. Note that, when constructing the boolean encoding of the graph, a single subgraph isomorphism test is performed, which is less expensive compared to the subgraph isomorphism counting required in the following step.

Once a flattened table is constructed, we perform the selection of subgraphs using existing feature selection methods. Specifically, we use $\mathcal{X}^2$ (Liu & Setiono, 1995), XGBoost (Chen & Guestrin, 2016), Mutual Information (Kraskov et al., 2004), and Random Forests (Breiman, 2001). While subgraph miners often already include a selection process, we find out that executing an additional selection step improved the performances (see Section 4).

### 3.3 COUNTING

As a result of the selection process, we are left with a set of subgraphs $S' = \{S'_1, S'_2, \ldots, S'_k\}$, with $k < m$. In this step, we finally exploit the generated substructures to augment the representation of the input graphs. Following the approach of GSN (), the goal is to count, in each input graph, how many times a node (resp. an edge, see Figure 3) appears in subgraphs that are isomorphic to the mined substructures. Then, such counts can be exploited in the representation of the input graphs or in the message passing of a graph neural network model. Specifically, given an input graph $G \in D$

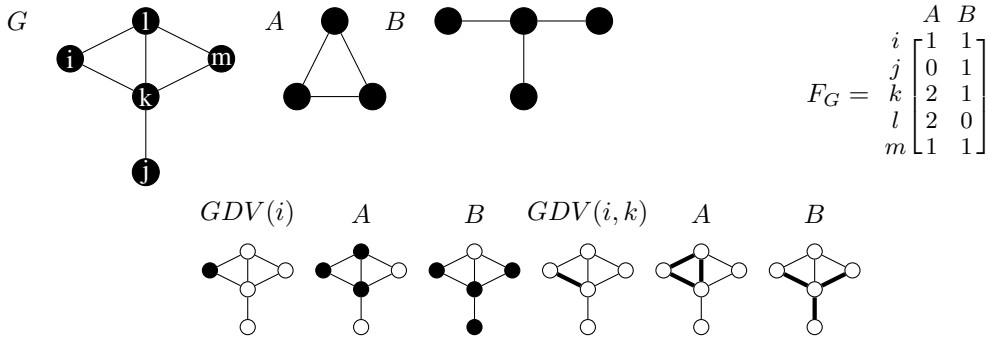

Figure 3: $F_G$ represents the constructed Graphlet Degree Vector (GDV) of all the nodes of graph $G$ for the subgraphs $A$ and $B$. Matches of the subgraphs that touch node $i$, and matches of the subgraphs that touch edge $(i, k)$ are shown at the bottom.

and a mined subgraph $S_i$, we find the set $\mathcal{H}_i$ of all the subgraphs $H_i$ of $G$ that are isomorphic to $S_i$, i.e. $\mathcal{H}_i = \{H_i : H_i \subseteq G, H_i \simeq S_i\}$.

We define the vertex structural feature $x_i(v)$ as follows:

$$x_i(v) = |\{H_i : H_i \subseteq G, H_i \simeq S_i, v \in N(H_i)\}| \tag{7}$$

i.e., how many subgraphs $H_i$ of $G$ and isomorphic to $S_i$ contain $v$. Similarly, we define the edge structural feature $x_i(v, u)$ as follows:

$$x_i(v, u) = |\{H_i : H_i \subseteq G, H_i \simeq S_i, (v, u) \in E(H_i)\}| \tag{8}$$

i.e., how many subgraphs $H_i$ of $G$ and isomorphic to $S_i$ contain $(v, u)$. The corresponding structural feature vectors are defined as $x(v) = [x_1(v), ..., x_k(v)]$ and $x(v, u) = [x_1(v, u), ..., x_k(v, u)]$

### 3.4 DISCUSSION

It is noteworthy that AUTOGSN involves computationally intensive operations, such as isomorphism testing and subgraph counting. Nonetheless, when compared with methodologies that incorporate analogous principles within the neural network architecture (Morris et al., 2019; Maron et al., 2019), two distinct advantages emerge. Firstly, by being employed as a preprocessing phase prior to the training procedure, these computations are applied only once, as opposed to being repeated during each forward pass of the network. Secondly, by performing such operations outside the computational graph of the network, we circumvent the constraints imposed by the tensor operations of deep learning frameworks like PYTORCH or TENSORFLOW and instead gain the flexibility to leverage specialised software tailored to each of these computational steps.

## 4 EXPERIMENTS

In this section, we evaluate empirically our proposed framework on different graph classification datasets. AUTOGSN is used to generate and select subgraphs, that are then encoded in *GSNs* (Bouritsas et al., 2022). For a fair comparison, we have reproduced the results using the TUDataset benchmark framework (Morris et al., 2020). We answer the following research questions:

- Q1. How does AUTOGSN perform on graph benchmarks?
- Q2. How does AUTOGSN qualitatively compare to hand-crafted features?
- Q3. Do the subgraph mining techniques improve the expressivity of GNNs?
- Q4. Which of the subgraph miners of AUTOGSN performs the best on average?

The following paragraphs illustrate our findings for each of these questions.

Table 1: Graph classification accuracy on TUD Dataset. Feature selection method are denoted with *RF* - Random Forest, *MI* - Mutual Information, $\mathcal{X}^2$, *XGB* - XGBoost. The best performing AutoGSN setting is reported for AutoGSN-v and AutoGSN-e.

| Dataset | MUTAG | PTC_MR | PROTEINS | NCI1 | IMDB-B | IMDB-M |
|---|---|---|---|---|---|---|
| GIN | $89.4 \pm 5.1$ | $64.5 \pm 2.5$ | $75.8 \pm 1.2$ | $83.4 \pm 2.7$ | $75.0 \pm 0.9$ | $51.6 \pm 0.8$ |
| GSN-v | $90.9 \pm 1.0$ | $65.2 \pm 1.2$ | $74.4 \pm 0.6$ | $82.5 \pm 0.3$ | $71.9 \pm 0.4$ | $46.5 \pm 0.9$ |
| GSN-e | $90.4 \pm 1.0$ | $64.9 \pm 1.6$ | $73.9 \pm 0.3$ | $83.3 \pm 0.5$ | $\mathbf{72.9 \pm 0.3}$ | $51.6 \pm 0.4$ |
| AutoGSN-v | $\mathbf{94.4 \pm 0.6}$ | $\mathbf{66.6 \pm 0.8}$ | $\mathbf{76.1 \pm 0.5}$ | $\mathbf{83.5 \pm 0.2}$ | $72.7 \pm 0.4$ | $51.0 \pm 0.3$ |
| Subgraphs | GSPAN RF | ANYBURL $\mathcal{X}^2$ | GSPAN MI | ANYBURL RF | RDFRULES RF | RDFRULES RF |
| AutoGSN-e | $93.5 \pm 1.0$ | $66.3 \pm 0.3$ | $75.4 \pm 0.4$ | $83.3 \pm 0.4$ | $72.7 \pm 0.2$ | $\mathbf{51.8 \pm 0.3}$ |
| Subgraphs | GSPAN MI | GASTON XGB | GSPAN $\mathcal{X}^2$ | ANYBURL MI | RDFRULES RF | RDFRULES RF |

Table 2: Set of AutoGSN features similar to human-designed ones. Feature selection method are denoted with *RF* - Random Forest, *MI* - Mutual Information, $\mathcal{X}^2$, *XGB* - XGBoost.

| Dataset | Human-Designed Features | AutoGSN | AutoGSN Setting |
|---|---|---|---|
| MUTAG | Cycles $k = [3, 12]$ | Cycles $k = \{5, 6, 9, 10, 11\}$ | ANYBURL & MI |
| PTC_MR | Cycle $k = [6]$ | Cycle $k = \{6\}$ | ANYBURL & XGB |
| PROTEINS | Cliques $k = [3, 4]$ | Clique $k = \{3\}$ | ANYBURL & $\mathcal{X}^2$ |
| NCI1 | Cycle $k = [3]$ | Cycle $k = \{3\}$ | RDFRULES & RF |
| IMDB-BINARY | Cliques $k = [3, 5]$ | Clique $k = \{3\}$ | RDFRULES & RF |
| IMDB-MULTI | Cliques $k = [3, 5]$ | Clique $k = \{3\}$ | RDFRULES & RF |

**A1: AutoGSN obtains state-of-the-art performance in the majority of the datasets (Table 1).**
We evaluate AutoGSN on datasets from the TUD benchmarks (Morris et al., 2020). We select six datasets from domains of bioinformatics and social networks. We compare AutoGSN with GIN (Xu et al., 2018) and GSNs (Bouritsas et al., 2022). We build upon the same evaluation protocol of (Xu et al., 2018). Specifically, we create 10-fold cross-validation sets and we average the corresponding accuracy curves. Since results are very dependent on weights initialisation, we perform 5 runs with different seeds for each fold, as suggested in (Morris et al., 2020). The corresponding 50 curves (10 runs x 5 seeds) are then averaged, obtaining an averaged validation curve. We select the single epoch that achieved the maximum averaged validation accuracy, and we report also the standard deviation over the 50 runs at the selected epoch. Table 1 lists datasets, best-performing subgraphs generated by AutoGSN used in our benchmark, and their respective accuracy on the test set.

**A2: Some of substructures extracted by AutoGSN match human-designed features.**
Human-designed features used in GSNs (Bouritsas et al., 2022) reach state-of-the-art results. Since we know these features, we can see if any of those features exist in generated features by AutoGSN. Table 2 shows the generated set of AutoGSN features that exist in the set of human-designed features for each dataset used in our experiments.

**A3: GNNs extended with AutoGSN features are more expressive than those using human-crafted features (Figure 4).** We perform an experiment on a synthetic dataset composed of 4152 strongly regular graphs obtained from (http://users.cecs.anu.edu.au/ bdm/data/graphs.html). All the graphs in the dataset are not isomorphic. In this experiment, we generate subgraph features with AutoGSN and, then, we use them to perform a single forward pass of a GSN architecture with random weights. By comparing the output embeddings, we are able to understand whether the network is able to discriminate the non-isomorphic graphs, which is a signal of their expressivity (Bouritsas et al., 2022). In Figure 4, we show the failure percentage with different classes of features, either human-designed (Bouritsas et al., 2022) or automatically extracted by AutoGSN. GSNs extended with automatically mined features and encoded in nodes (AutoGSN-v) manage to distinguish almost as many graphs as GSNs with human-crafted features. This support our claim that, with no

human intervention, AUTOGSN is still able to highly increase the expressivity of GNNs, with no computational increase in the forward pass, but only as a preprocessing counting step.

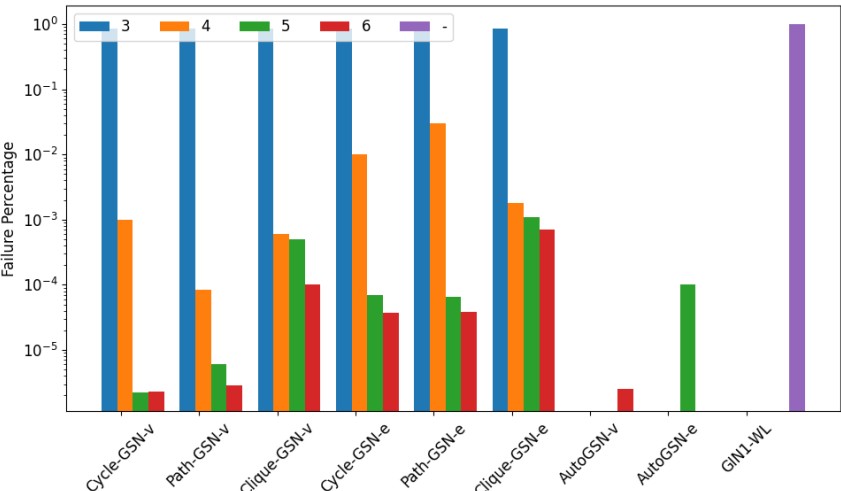

Figure 4: Strongly regular graphs isomorphism test (log scale, smaller values are better). Different colours indicate different substructure sizes.

**A4: Rule miners tend to extract better substructures than subgraph mining approaches.** While there is no overall best performing miner, as showed in Table 1, rule miners tend to provide the better features on average. This is particularly evident in large datasets, where only rule miners are able to extract relevant features.

## 5 RELATED WORK

To extract graph substructure and make use of them for downstream tasks, multiple approaches exist. We divide them into three groups: 1) graph neural networks and substructures, 2) subgraph mining techniques, and 3) rule mining on knowledge graphs. The methods described in this section are used throughout our study, we make use of subgraph mining and knowledge graph completion techniques with GNNs.

**Graph Neural Networks and Substructures** Since the limitations of MPNNs have been uncovered, there has been a substantial amount of work investigating GNNs that are capable of capturing more topological information. This direction of research is also known as *Beyond Weisfeiler-Lehman* since the basic idea is to increase the expressivity of GNNs beyond the *1-WL* isomorphism test. These GNNs are also known as substructure-based GNNs and they can be divided into two groups: 1) the first group improves MPNNs by injecting subgraph information into the aggregation process (similar to GDV, see Figure 3) and 2) the second group decomposes the graph into a few subgraphs and then merges their embeddings to produce an embedding of the entire graph.

Substructure-based GNN such as GNN-AK (Zhao et al., 2021) takes an approach to avoid the expensive process of constructing GDVs (see Figure 3). Instead for each node $v$ a subgraph $G_v$ is sampled from its neighbours and then encoded by applying a base MPNN to obtain a final node embedding. NGNN (Zhang & Li, 2021) creates node vectors of features in a similar way as GNN-AK. Specifically, it extracts nodes and edges in the 1-hop neighbourhood of a node $v$ as a neighbourhood subgraph $G_v$ that has to be encoded by a GNN.

GNN architectures based on graph decomposition take another approach when it comes to making use of rich substructure information. For example, SUGAR (Sun et al., 2021) uses reinforcement learning to select discriminative subgraphs. Over the selected subgraphs, embeddings are learned using a GNN, and as a last step, a *readout* function is applied over embeddings that can serve for graph classification.

SubGNN (Alsentzer et al., 2020), decomposes subgraphs and treats them as individual graphs. The problem changes to predicting subgraph properties $C$ (independent labels) by learning subgraph representations that recognise and disentangle the heterogeneous properties of subgraphs and how they relate to underlying graph $G$.

MPSN (Bodnar et al., 2021b) is another model that performs message passing on simplicial complexes. One can see 0-simplices as vertices, 1-simplices as edges, 2-simplices as triangles, and so on. The authors define four types of adjacencies to define adjacent simplices: 1) Boundary simplices is an edge between its vertices; 2) The co-boundary simplices of a vertex are given by the edges that they belong; 3) The lower-adjacent edges are given by the common line-graph adjacencies; 4) Upper adjacencies between vertices give the regular graph adjacencies. In other words, the defined topological properties are fixed and used as "extra" information in message-passing, thus being capable of passing more structural information. MPSNs are constrained by the combinatorial substructure of simplicial complexes. CIN (Bodnar et al., 2021a) extends MPSNs and operates on cell complexes, thus generalises and subsumes the MPSNs. Cell complexes are topological spaces that are constructed by combining simple primitives called cells.

**Subgraph Mining Techniques** The goal of subgraph mining algorithms is to identify an optimal set of subgraphs. These algorithms can be divided into two categories based on the type of supervision that they use. The first category is referred to as Frequent Subgraph Mining (FSM), where no external supervision is used. The second category, known as Discriminative Frequent Subgraph Mining (DFSM), uses supervised or semi-supervised methods for subgraph mining. Frequent Subgraph Mining techniques (Yan & Han, 2002; Nijssen & Kok, 2005; Ketkar et al., 2005; Ribeiro et al., 2021) are focused on efficiently finding subgraphs from a set of graphs or a knowledge graph with a frequency no less than a given support threshold. The limitation of these methods is that the graph pattern search space grows exponentially with the pattern size (Cheng et al., 2014). On the other hand, Discriminative Frequent Subgraph Mining (Saigo et al., 2009; Jin et al., 2010) techniques search for patterns (subgraphs) from a set of all possible subgraphs based on the label (target) information. The problem of DFSM can be defined as follows: Given a graph database $D = \{G_1, G_2, \ldots, G_n\}$ and an objective function $\mathcal{F}$, find all subgraphs $g$ such that $\mathcal{F}(g) \geq \delta$ where $\delta$ is a significance threshold; or alternatively find the $k$-best subgraph $g'$ such that $g' = \arg\max \mathcal{F}(g)$ (Cheng et al., 2014).

While FSM and DFSM techniques have shown promise in mining discriminative frequent subgraphs from graph datasets, they also have limitations in terms of computational complexity, incomplete discriminative features, handling data heterogeneity, scalability to graph size, imbalanced class distribution, and scalability to multiple classes (Welke et al., 2018; Welke, 2020).

**Rule Mining on Knowledge Graphs** Rule mining on knowledge graphs is a data-driven approach used to discover patterns or subgraphs within large graphs. This method involves predicting missing nodes or edges in the knowledge graph. One key advantage of knowledge graph rule mining techniques is their scalability, as they primarily focus on closed and connected rules or subgraphs. Additionally, some approaches, such as AMIE+ (Galárraga et al., 2015), utilise predefined metrics such as confidence and support (similar to FSM), to speed up the mining process. Some of the well-known knowledge graph completion methods are ANYBURL (Meilicke et al., 2019) and RDFRULES (Zeman et al., 2021).

## 6 CONCLUSION

In this paper, we introduce AUTOGSN, a framework for enhancing the expressiveness of Graph Neural Networks by incorporating complex structural information automatically obtained through subgraph mining techniques.

The results of our experiments demonstrate that AUTOGSN can achieve performance comparable to, or even superior to, GNNs relying on manually crafted features. This finding not only opens up new possibilities for improving the effectiveness of GNNs but also alleviates the burden of requiring specialised domain knowledge and significant human effort to engineer features.

In the future, we plan to extend our framework with more advanced hyperparameter tuning techniques, moving towards a fully automated machine learning pipeline. Moreover, we will investigate how to more closely integrate the mining component and neural components following a neurosymbolic approach.

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
