# OpenReview forum: "Subgraph Mining for Graph Neural Networks"
_ICLR.cc/2024/Conference — ICLR 2024 Conference Withdrawn Submission_

### Official Review · Reviewer_x5SK · 2023-10-15

**Soundness:** 3 good
**Presentation:** 3 good
**Contribution:** 3 good
**Rating:** 6
**Confidence:** 3

**Summary:**

The paper deals with message-passing graph neural nwteorks (MPGNNs) that extend node vectors with information about subgraph counting at such a node. These "node-enriched" MPGNNs lie between 1WL and higher-order WL in terms of expressive power. Depending on the patterns used in subgraph counts, they extend WL but do not suffer from the performance issues of higher order GNNs.

While node-enriched MPGNNs have been studied both theoretically and practically, the patterns that are to be counted at each node are often selected by human inspection. This is clearly non-optimal. The paper presents a framework, based on data mining techniques, that assists in the automatic selection of such patterns. Several experiments are presented evaluating the suitability of the proposal for different graph classification problems.

**Strengths:**

- The paper deals with a concrete practical problem of interest to the ICLR audience, and that, to the best of my knowledge, had not been handled to date.
- The presentation of the paper is very good.
- The empirical results confirm that the proposed approach is meaningful; it is often not worst, and sometimes even better than the human-based approach.

**Weaknesses:**

- The novelty of the paper is rather limited: node-enriched MPGNNs have bee studied before and the techniques used for selecting the node features come from the data mining literature. The contribution really lies on the combination of these ideas, which is certainly worthy, but does not suffiice for a stronger accept in my view.

**Questions:**

- Node-enriched MPGNNs have not only been studied by Bouritsas et al., but also in several other papers; e.g.,

Pablo Barceló, Floris Geerts, Juan L. Reutter, Maksimilian Ryschkov:
Graph Neural Networks with Local Graph Parameters. NeurIPS 2021: 25280-25293

You should include this and other papers in your reference list.

- Page 2: "To address these challenges": Which challenges?

- Page 2: Whats is the difference between your contributions listed as 1) and 2)

- Page 3: "Details on how to choose ...": I think better reference can be found here

- Page 3: The current agreement is that the name is Leman, not Lehman.

- Page 3: Please provide citations for k-WL.

- Page 5: Following the approach of GSN: missing citation

- Page 7: All the graphs in the dataset are not isomorphic --> better rephrase as "No pair of different graphs in the dataset are isomorphic"

- Page 7: This supportS

- Page 9: Subgraph mining techniques: You claim that the limitation of some of the subgraph mining techniques is that they suffer from an exponential search space. Why and how your method avoids this?

---

### Official Review · Reviewer_8yY4 · 2023-10-18

**Soundness:** 3 good
**Presentation:** 2 fair
**Contribution:** 1 poor
**Rating:** 3
**Confidence:** 4

**Summary:**

This paper introduces an automated framework that combines traditional graph and deep learning. By using subgraph mining techniques, the framework produces substructures that bolster the effectiveness of substructure-based MPNNs. The approach utilizes multiple subgraph mining methods to produce potential performance-enhancing substructures. Different feature selection strategies are then used to pinpoint the most beneficial substructures, which are subsequently encoded at the node or edge level of the graphs.

**Strengths:**

The authors provided a detailed introduction to the problem definition related to the problem.

**Weaknesses:**

1. The paper's novelty appears constrained, as it primarily amalgamates concepts from prior works without offering substantial contributions or fresh insights.
2. Some highly related work should be cited and discussed.
* Yu, X., Liu, Z., Fang, Y., & Zhang, X. (2023). Learning to Count Isomorphisms with Graph Neural Networks. Proceedings of the AAAI Conference on Artificial Intelligence, 37(4), 4845-4853.
* Liu, Xin, et al. "Neural subgraph isomorphism counting." Proceedings of the 26th ACM SIGKDD International Conference on Knowledge Discovery & Data Mining. 2020.
* Chen, Zhengdao, et al. "Can graph neural networks count substructures?." Advances in neural information processing systems 33 (2020): 10383-10395.
3. The authors should provide the hyperparameter settings used for the proposed model and the baselines in the paper.

**Questions:**

See weaknesses.

---

### Official Review · Reviewer_qGJB · 2023-10-31

**Soundness:** 2 fair
**Presentation:** 2 fair
**Contribution:** 1 poor
**Rating:** 3
**Confidence:** 3

**Summary:**

This paper explores the use of subgraph mining and feature selection techniques to enhance the expressiveness of Graph Neural Networks (GNNs) for graph learning tasks. The AUTOGSN framework is proposed to increase the expressivity of GNNs by augmenting the feature representation of input graphs with automatically discovered subgraph patterns. The authors demonstrate that automatically extracting structural information through subgraph mining can lead to comparable or even superior performance compared to GNNs relying on manually crafted features.

**Strengths:**

1. This paper proposes an automated framework named AUTOGSN that bridges the gap between traditional graph learning and deep learning by leveraging subgraph mining techniques to generate substructures that enhance the performance of substructure-based Graph Neural Networks (GNNs).
2. The framework selects various off-the-shelf subgraph mining techniques and employs different feature selection strategies to identify the most useful substructures. The paper also provides an extensive evaluation on graph classification tasks.

**Weaknesses:**

1. Very limited contribution. The AUTOGSN framework is ground on the GSN [1] framework with off-the-shelf subgraph mining techniques, including frequent subgraph mining techniques, discriminative subgraph mining techniques and rule mining techniques.
2. Very limited empirical analysis. The main experiments of graph classification were conducted on five graph datasets and the proposed AUTOGSN was compared to only GIN and GSN, which makes the evaluation insufficient. Further, there lacks necessary experiments like ablation studies, parameter sensitivity analysis, etc.
3. Too much related works. Introduction to related works and preliminaries in Chapter 2, Section 3.1, Section 3.2, and Chapter 5 takes up about 4.5 pages (50%) of the whole paper, which greatly damages the role of proposed AUTOGSN.

[1] Improving graph neural network expressivity via subgraph isomorphism counting, TPAMI 2022.

**Questions:**

1. Compared to other state-of-the-art graph classification baselines, what is the advantages of AUTOGSN? It would be great if the authors could provide some extensive results.
2. Computational complexity analysis, sufficient analysis and expressive ability analysis of AUTOGSN are not discussed in the main contents.
3. There is a trade-off between complexity and expressivity in GNNs, where standard GNNs are only as expressive as the 1-WL isomorphism test. How to consider the trade-offs between complexity, expressivity, and interpretability in AUTOGSN?
4. In addition to subgraph mining and subgraph selection, which are mainly built upon off-the-shelf techniques, what are the main innovations and contributions of AUTOGSN?

---

### Official Review · Reviewer_nYJy · 2023-11-01

**Soundness:** 3 good
**Presentation:** 4 excellent
**Contribution:** 2 fair
**Rating:** 3
**Confidence:** 5

**Summary:**

The paper introduces AutoGSN, an approach for automatically inferring important subgraph structures using traditional subgraph mining techniques and subsequently incorporating this information within the message passing of graph neural networks (GNNs). To this end, AutoGSN builds on the GSN model, which itself manually selects important subgraphs and computes additional node and edge features by counting the number of such subgraphs that nodes (resp. edges) belong to. More concretely, AutoGSN extends GSN by replacing manual subgraph selection with an automated pipeline, consisting of three steps. First, AutoGSN calls a suite of sub-graph mining tools to identify relevant sub-graph structures based on different criteria (frequency, discriminability, rule mining). Second, AutoGSN selects the most important subgraphs by i) computing Boolean flags for each candidate subgraph, indicating its existence within the input graphs (and thus avoiding the expensive GSN subgraph counting step in favor of a more tractable isomorphism check), and ii) using feature selection techniques, e.g., XGBoost, to select the most informative features. Third, AutoGSN proceeds analogously to GSN and computes the counts of all subgraphs for inputs graphs' nodes/edges to produce a feature set that supplements standard GNN message passing.

Empirically, AutoGSN is evaluated on the TUDataset benchmarks and shown to yield very good results, matching and at times surpassing manual subgraph selection (GSN). Moreover, the AutoGSN subgraph selection procedure is also shown to select similar important subgraph in some cases as manual selection, and the subgraphs selected help improve model expressiveness at least as much as the latter (with this being validated empirically on a dataset of strongly regular graphs). Finally, the paper briefly discusses the choice of subgraph mining techniques, and highlights that, though no individual approach is a clear winner, rule mining techniques are the better performer on average.

**Strengths:**

- The idea of automating subgraph selection is simple, intuitive, and easy to justify.
- The presentation of the approach is very clear
- The analysis of selected subgraphs in the empirical evaluation section is interesting

**Weaknesses:**

- The experimental evaluation is very limited and is missing several important details. First, the paper mentions that the subgraph mining operations incur a pre-computation cost. However, this is not substantiated with concrete runtimes / complexity analyses for the concrete model instantiation. Even if the automatic subgraph mining is a pre-computation, it is very important to establish the feasibility of this operation, particularly as this pre-computation must run on each dataset and may in some cases involve prohibitive scaling behaviors that outweigh any potential performance improvements. Moreover, the experimental analysis does not include other baselines from subgraph GNNs to better locate the relative merit of the performance gains from AutoGSN. Indeed, It is not surprising that AutoGSN would surpass GSN, but it  would be much more compelling if it is competitive / surpasses several SOTA models (including other "beyond 1-WL approaches")), e.g., ESAN [1], Graphormer [2], etc. Third, it is important to also validate the value of this approach on more datasets (e.g., OGB), in conjunction with other baselines as mentioned earlier, to further justify the need for this more involved pre-computation.

- The contribution of this paper in its current form  is also very limited. Indeed, the main idea in this paper is to apply automated subgraph mining to graph datasets and subsequently plug this into GSN. This is itself is not a novel idea: feature selection is a common empirical step in practice, and applying such techniques on the outputs of multiple subgraph mining solutions is relatively straightforward.  Hence, the main avenue for contribution lies in the insights extracted from the empirical analysis. However, the current empirical analysis does not provide many compelling insights: The current expressiveness analysis / subgraph discussions are interesting and show, e.g., that automated subgraph selection can align well with manual selection, but does not provide any rigorous analysis on more pressing questions. In particular, it is not clear how/why certain subgraphs succeed/fail on datasets and how their impact varies relative to the dataset objective/task. It is also completely open whether the current approach (subgraph mining) is unlocking performance levels that cannot be matched by alternative approaches (as mentioned in the first point). Third, the paper does not produce deeper insights on expressiveness and the role of subgraphs there: It is obvious that, in the limit, selecting the right subgraphs can unlock full expressiveness on a given dataset. However, it is totally unclear whether the subgraphs that allow for improved expressiveness are the same that lead to improved task performance (i.e., do the two objectives compete when selecting subgraphs?), and this paper has an opportunity to address this point. I therefore highly recommend that the authors make a deeper dive in their analysis along these lines.

[1] Bevilacqua et al.  Equivariant subgraph aggregation networks. ICLR 2022.
[2] Ying et al. Do transformers really perform badly for graph representation? NeurIPS 2021.

**Questions:**

No direct questions at this point. Instead, please simply address all suggestions / comments I provided in the weaknesses section above.